# Numerical Study of the Influence of Geometric Features of Dimple Texture on Hydrodynamic Pressure Generation

**Yuan Wei *** , **Robert Tomkowski** and **Andreas Archenti**

Manufacturing and Metrology Systems, KTH Royal Institute of Technology, 10044 Stockholm, Sweden; rtom@kth.se (R.T.); archenti@kth.se (A.A.)

**\*** Correspondence: yuanwei@kth.se; Tel.: +46-8-790-83-53

**Abstract:** Controlling friction and wear are essential for reducing energy loss and lengthening the life span of friction pairs in sliding contacts. Surface texturing is an effective and efficient way to reduce friction and wear, especially under lubricated conditions. Dimple texture on friction pair surfaces has been verified to enhance the lubrication condition and shift the lubrication regime by generating additional hydrodynamic pressure. However, the geometric features and distribution of the microstructures considerably influence the production of additional pressure. Choosing and designing dimple texture with suitable geometric features is necessary and important for texturing applications. In this study, computational fluid dynamics (CFD) are used to investigate the effect of the geometric features of dimple textures on pressure build-up. The influence of dimple shapes, dimple depths, minimum film thickness, dimple densities, and dimple surface angles are presented and discussed. Notably, the influence of the dimple surface angle is introduced and presented.

**Keywords:** surface texture; dimple; computational fluid dynamics; hydrodynamic pressure; geometric features

---

## 1. Introduction

Friction and wear control are important for sustainable development in society because most mechanical systems failure and energy loss are caused by friction and wear [1]. Many efforts have been made to reduce or minimize friction in mechanical systems. Surface texture is considered an efficient method to control friction and wear under lubricated conditions [2,3]. The tribology performance of modified surfaces with dimples and grooves has been investigated theoretically and experimentally over the last six decades. However, the texture effect was observed only under strict conditions, and it highly depends on the contact type, lubricant, motion of the interface, etc. In particular, the geometric shapes and dimensions of the texture pattern play a vital role in texture performance [4].

Under a lubricated condition, the dominant mechanism of the texture effect is the generation of additional hydrodynamic pressure. Computational fluid dynamics (CFD) has been introduced and is considered a powerful and efficient tool to investigate the texture effect under lubricated conditions [5,6]. CFD precisely predicts and simulates hydrodynamic pressure generation due to texture [5]. Dimple texture on surfaces has been investigated by many researchers. In 1966, Hamilton et al. enhanced lubrication on the face of a rotary shaft seal by applying micro-irregularities, which could be considered as dimple texture [7]. Hamilton's research opens the door to improving lubrication with surface texture. Since Hamilton's research in 1966, hundreds of theoretical and experimental studies on the surface texture effect and mechanism have been published. Rather than conducting experimental research, with the development of highly efficient algorithms and computational power,

CFD has improved as a theoretical, highly efficient and reliable way to investigate the surface texture effect. Arghir et al. [8] investigated pressure build-up by single dimple texture using CFD; they used two-dimensional and three-dimensional models to represent the dimple texture with different bottom profiles. Nanbu et al. [4] analyzed the dimple surface effect with different bottom shapes of dimples numerically. The results suggested the dimple bottom shape affected the film thickness. Papadopoulos et al. [9] performed simulations using commercial CFD software and found a higher load capacity when a squared dimple surface was applied. Zhang et al. [10] performed three-dimensional CFD simulations for water-lubricated step thrust bearings with different bearing dimensions, film thicknesses, and step heights and positions. They discovered optimized geometry parameters for step thrust bearings. Jiang et al. [11] investigated the film stiffness of dry gas seal with different shapes of groove using CFD, a higher film stiffness was observed with a dry gas seal which has spiral grooves. Shimizu et al. [12] performed two-dimensional CFD simulations for analyzing the inflow behavior of the lubricant to the inside of dimples with different size. Different flow-in behavior of two-phase flow are discovered regarding different dimple's dimensions.

However, in most studies, a simplified three-dimensional texture model, and sometimes a two-dimensional model, are used. Normally, dimples are modeled as regular cylinders or rectangles in the fluid domain. In this case, the geometric features that may influence the texture effects are neglected by the simplification. During the most commonly implemented texturing processes, including laser texturing, micromachining texturing, and micro-abrasive blasted texturing, the dimples are not regular shapes. The regular cylinder shape model is oversimplified, which may mislead the texture effect prediction when using a numerical method to make the prediction. Considering meshing and computational efficiency, importing the real surface for CFD simulation is improper as well. In this work, the dimples are mimicked by a three-dimensional model that considers the dimple surface angle, thereby trying to represent as accurately as possible the main geometric features of dimple textures.

In this paper, a comprehensive design of dimple texture and the geometric features of the dimples is performed and discussed. The influence of dimple geometric features in journal bearings like contact are investigated using CFD-based simulations. In addition, the dimple texture effects are compared using different dimple shapes and densities. The relationship between dimple depths and minimum film thickness is studied as well. A novel parameter, dimple surface angle, is introduced, and the effect of the dimple surface angle is determined. The objective of this study is to provide a better understanding of the role of dimple geometric features in the dimple texture effect. References for the dimple texture design are presented for further applications. This study only considers convergence contact, and the simulation is modeled and simplified based on journal bearings.

## 2. Modeling Dimples

In present work, three-dimensional CFD simulations are performed using the partial flow domain model based on the journal bearings like contact condition. Different dimple textures are built on the models. The influence of dimple geometric features on pressure built-up are discovered by transient pressure-based CFD simulation.

The CFD model is built based on a journal bearing contact condition. Figure 1 shows a schematic of the journal bearing under hydrodynamic lubrication. The model geometric features are calculated based on a journal bearing with $D_b = 80$ mm and $d_b = 79.95$ mm. A small flow area is picked from the journal bearing contact to build the CFD model. The curved convergence flow area is represented by the model with a sloped top surface, as shown in Figure 1. The three-dimensional CFD model is shown in Figure 2. Dimples are modelled on the bottom surface of the model. A minimum film thickness $H_{min}$ is set in advance, and a top surface slope angle $\beta$ and inlet film thickness $H_{in}$ are calculated directly based on $D_b$, $d_b$, and $H_{min}$. For comparison, a non-textured model is included as well. The textured area is on the bottom surface of the model, and the top wall is set with translational motion. The flow direction is also shown.

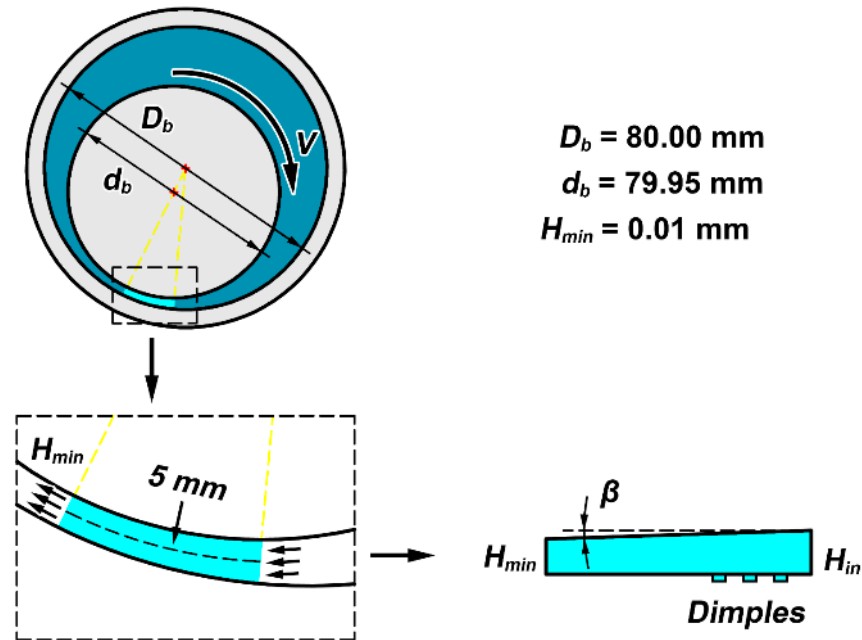

**Figure 1.** Schematic of the journal bearing contact under hydrodynamic lubrication (5 mm indicates the length of the picked flow area).

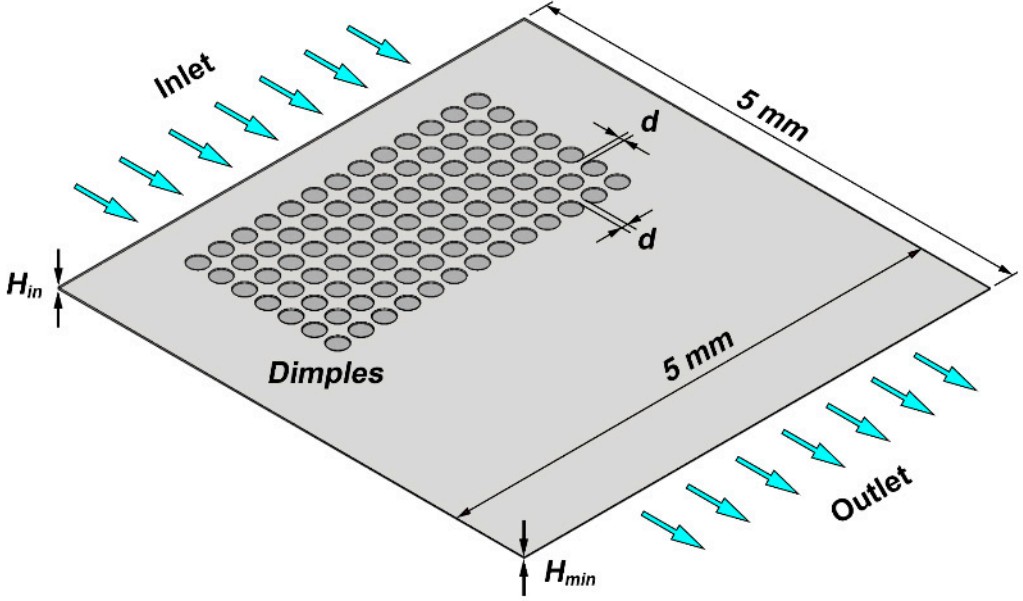

**Figure 2.** Schematic of the journal bearing contact under hydrodynamic lubrication in the computational fluid dynamics (CFD) model with the dimples on the bottom surface.

Dimples with different shapes, depths, and surface angles are modeled in this present work. The dimple surface angle is introduced to mimic a real dimple profile. Figure 3 shows a comparison of a real dimple profile, conventional dimple model and the model profile with a dimple surface angle. Compared to the convential dimple model, the proposed model that with the dimple surface angle is more close to the real dimple profile. Thus, it is important to understand whether dimple surface angle plays a role in the simulation of texture effect and how it plays that role.

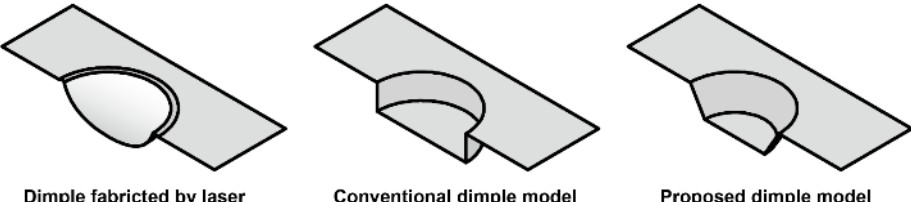

**Figure 3.** Images of dimples fabricated by a laser texturing process on steel, drawn based on the published information [12], the conventional model, and proposed dimple model.

Five basic dimple shapes are designed and modeled, including a circle, square, ellipse, trapezoid and triangle. A rotated ellipse, trapezoid, and triangle are included, as shown in Figure 4, to consider the influence of a shape's orientation. The main dimensions of the dimples are designed as similar as possible to make them more comparable. Dimples are distributed on a surface uniformly, with a certain distance *d* between the dimples (Figure 2).

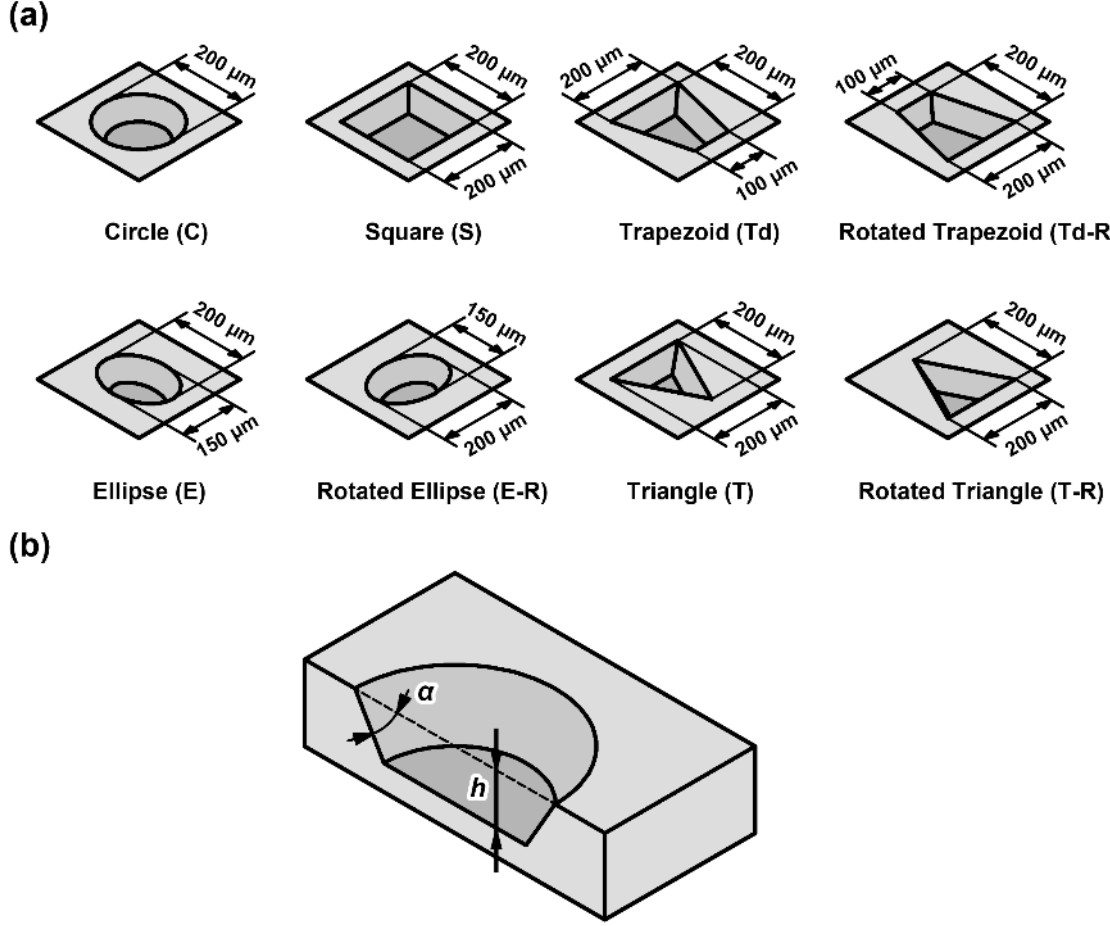

**Figure 4.** (**a**) Designs of the dimples with different shapes and (**b**) the definition of dimple surface angle *α* and dimple depth *h*.

Investigation of the influence of five variables are performed. Besides the dimple shapes, the dimple depths, dimple number, dimple surface angle and film thickness are considered in the simulations as well. The details of simulation case condition regarding the five variables. are listed in Table 1. Four different cases are investigated by four sets of simulations respectively. In Table 1, the case number also indicates the subsection which the results of that case are shown in Section 4. The effect of dimple shapes is investigated in Case 4.1. The dimple number's influence is examined

in Case 4.2. The influence of dimple depth and minimum film thickness is presented in Case 4.3. The dimple surface angle effect is introduced in Case 4.4.

**Table 1.** Dimple features and film thicknesses for different simulation cases.

| Case | Dimple Shape | Dimple Depth | Dimple Number | Dimple Surface Angle | Film Thickness |
|------|--------------|--------------|---------------|----------------------|----------------|
| 4.1 | Circle, square, etc. | 10 µm | 91 | 60° | 10 µm |
| 4.2 | Circle | 10 µm | 15~91 | 60° | 10 µm |
| 4.3 | Circle | 3~40 µm | 91 | 60° | 10~30 µm |
| 4.4 | Circle | 10 µm | 91 | 90~7.5° | 10 µm |

## 3. Numerical Modeling

The method of CFD modeling and simulation is described. The governing equations and settings of the simulation are introduced. Boundary conditions applied in the simulation are also introduced.

The CFD code ANSYS FLUENT is used in the present study. The model meshes with refinement at the textured area. The total element number is around $1.2 \times 10^6$ for a single model. The mass conservation equation and momentum conservation equation need to be solved.

$$\nabla \cdot \mathbf{v} = 0 \tag{1}$$

$$\rho(\mathbf{v} \cdot \nabla)\mathbf{v} = -\nabla p + \mu \cdot \nabla^2 \mathbf{v} \tag{2}$$

Cavitation is considered as a vital source for additional hydrodynamic pressure due to the texture [2], thus, cavitation model is used. The flow is defined as two-phase flow, water and water vapour. Water lubricated bearings have been used in many applications as an environment friendly component [13]. In addition, the viscosity of water at room temperature is slightly lower than most lubricant's viscosity at working temperature (e.g. low viscosity engine oil (0W20) used in truck engines at 100 °C). Therefore, water is a reasonable lubricant choice for general numerical studies regarding the texture effect under a lubricated condition.

The properties of water and water vapour at room temperature are listed in Table 2. The mass transfer mechanism between two phases is defined using the Schnerr and Sauer model, which provides stable computing and a solution that is less sensitive to mesh density for considering the cavitation in ANSYS FLUENT [14].

**Table 2.** Physical properties of water and water vapour at room temperature.

| Physical Properties | Value | Unit |
|---------------------|-------|------|
| Density of water | 998.2 | kg/m$^3$ |
| Dynamic viscosity of water | $1.003 \times 10^{-3}$ | kg/m·s |
| Saturated water pressure | 3540 | Pa |
| Density of water vapour | 0.554 | kg/m$^3$ |
| Dynamic viscosity of water vapour | $1.34 \times 10^{-5}$ | kg/m·s |

The governing equation for mass transfer between two phases in cavitation is the vapour transport equation:

$$\frac{\partial}{\partial t}(\alpha_v \rho_v) + \nabla \cdot (\alpha_v \rho_v \mathbf{v}) = R_e - R_c \tag{3}$$

where $R_e$ and $R_c$ account for the mass transfer between the two phases in cavitation, the $\alpha_v$ is the vapour volume fraction, $\rho_v$ is the vapour density, $\mathbf{v}$ is the vapour phase velocity. In the Schnerr and Sauer model, the mass transfer sources are defined as follows:

If $p \leq p_v$, then

$$R_e = \frac{\rho_l \rho_v}{\rho} \cdot \alpha_v (1 - \alpha_v) \cdot \frac{3}{R_b} \cdot \sqrt{\frac{2}{3} \cdot \frac{p_v - p}{\rho_l}} \tag{4}$$

If $p > p_v$, then

$$R_c = \frac{\rho_l \rho_v}{\rho} \cdot \alpha_v (1 - \alpha_v) \cdot \frac{3}{R_b} \cdot \sqrt{\frac{2}{3} \cdot \frac{p - p_v}{\rho_l}} \tag{5}$$

where the $R_b$ is the bubble radius, $\rho_l$ is the liquid density, and $p_v$ is the saturated liquid pressure.

Pressure based inlets and outlets are defined; the operating pressure is set to 101,325 Pa. The top wall is set with translational motion at 12 m/s. The bottom wall and two side walls are stationary. A no-slip condition is imposed on the walls.

The following assumptions are made in the present study:

1. The flow is considered as laminar, incompressible, and isothermal.
2. The influence of a solid structure formation due to the flow pressure is negligible.
3. Film thickness change due to the texture is negligible.

## 4. Results and Discussion

The simulations are performed with different dimple shapes, dimple depths, minimum film thickness, dimple densities, and dimple surface angles. The dimple features employed in each section are shown in Table 1. The results are plotted and discussed to describe the influence of those variables.

### 4.1. Effect of Dimple Shape

Dimples with different shapes are distrusted uniformly with constant density in 8 models (Case 4.1 as shown in Table 1). The pressure distributions on the bottom surface of the model for the different dimple shapes are shown in Figure 5. The square dimples, see Figure 5b, show the highest pressure build-up compared to the other shapes. The trapezoid, rotated trapezoid, and circle dimples provide pressure increases. The average pressures on the top surface of the model for different dimple shapes are plotted in Figure 6; this figure also lists the areas of each dimple shape. The average pressure for the non-textured model is added for comparison. Compared to the non-textured case, dimples of the 8 shapes which presented in this study could generate additional pressure. For the dimple shape influence, the average pressure on the top surface shows the same trend as the pressure distributions of the various shapes in Figure 5. The red dots in Figure 6 indicate the total area of dimples for each kind of the dimple shape. The "NT" in the x-axis indicates the non-textured case. The pressure change is in good agreement with the change in a dimple's area; the results indicate that as a dimple's area increases, more pressure build up is generated. In addition, by comparing the normally oriented shape with its associated rotated shape (e.g., ellipse compared to the rotated ellipse), we can see that the flow direction affects pressure generation as well.

By considering pressure generation, the square dimples should be the optimal solution for the surface texture with dimples. Several published numerical surface texturing studies use the square dimples. However, if we aim to utilize surface texturing in large-scale, industrial applications, the texturing efficiency and cost should be considered as well. If efficiency and cost are considered, the circle dimple is the most suitable candidate for further study because, compared to the other dimple shapes, circle dimples can be fabricated using many texturing processes, including micromachining, and on large-scale surfaces easily and quickly. It is foreseeable that circle dimples have more potential to be applied in industry compared to other shapes.

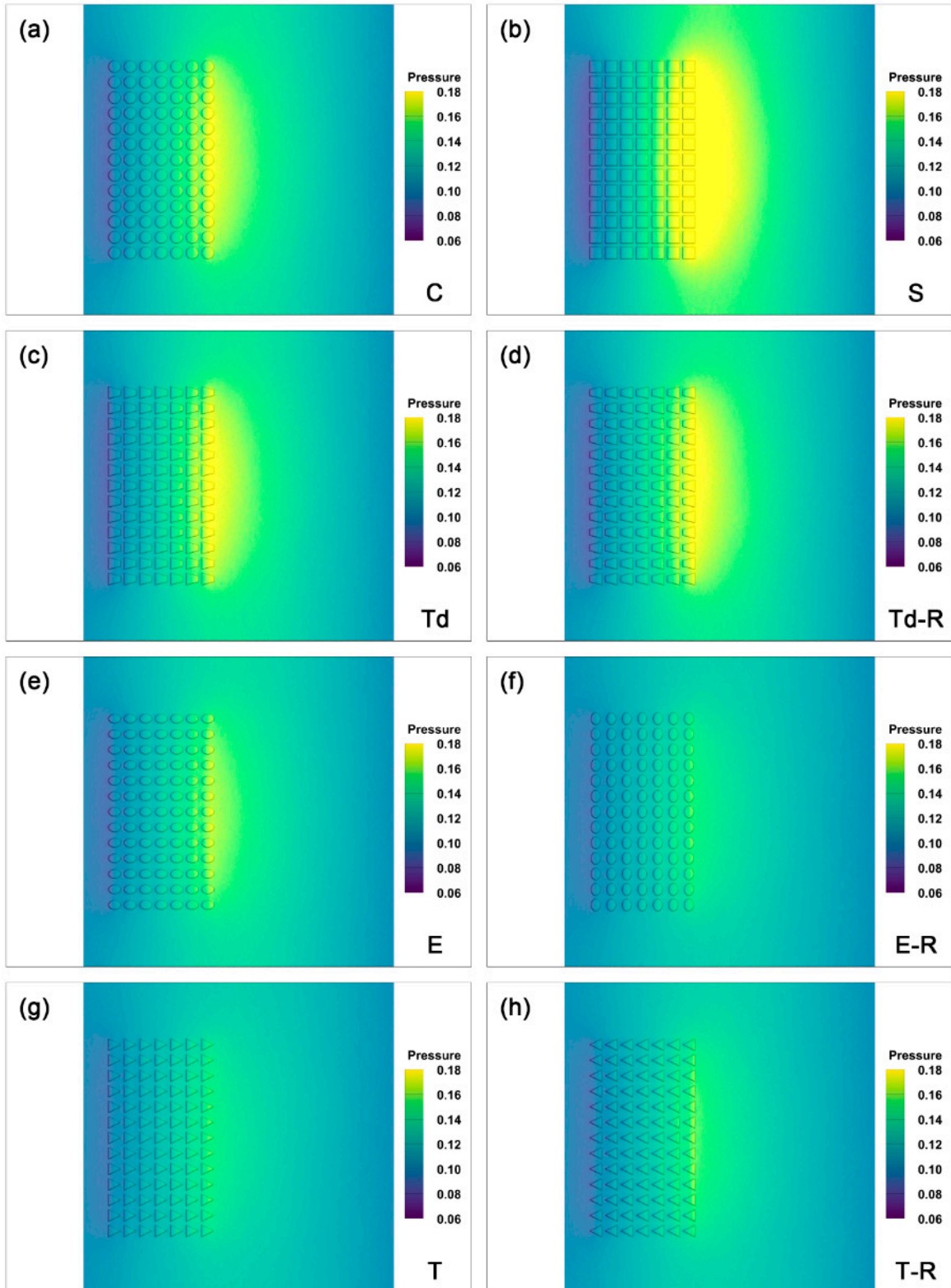

**Figure 5.** Pressure distributions on the bottom surface of the model for the circle (**a**), square (**b**), ellipse (**c**), rotated ellipse (**d**), trapezoid (**e**), rotated trapezoid (**f**), triangle (**g**), and rotated triangle (**h**).

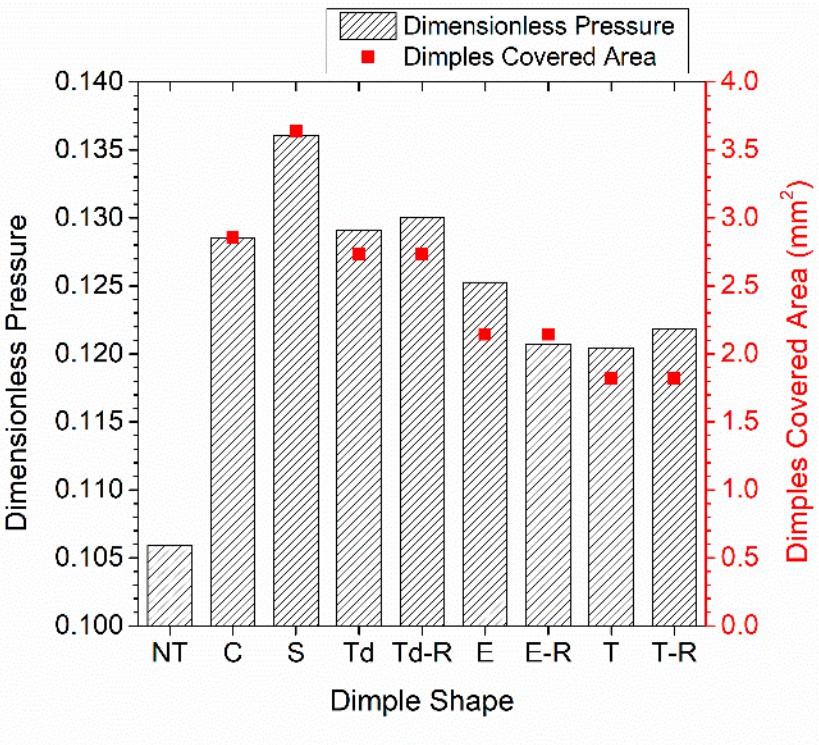

**Figure 6.** Comparison of average pressure on the top surface of the model for the different dimple shapes: non-textured (NT), circle (C), square (S), ellipse (E), rotated ellipse (E-R), trapezoid (Td), rotated trapezoid (Td-R), triangle (T), rotated triangle (T-R).

## 4.2. Effect of Dimple Density

Dimple density is investigated to determine its influence on hydrodynamic pressure generation Case 4.2 as shown in Table 1). Five simulations with different dimple densities are performed, as shown in Figure 7. Fifteen, 28, 45, 66, and 91 dimples are distributed over the same area, and the distance between dimples are 750 μm, 500 μm, 375 μm, 300 μm and 250 μm respectively. The results show that pressure is proportional to dimple density.

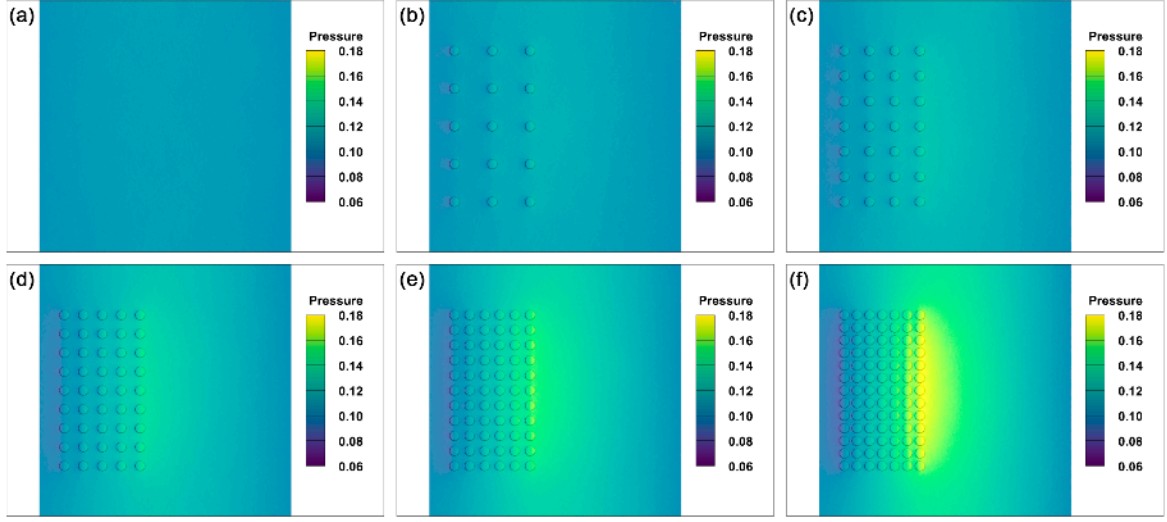

**Figure 7.** Pressure distributions on the top surface of the model for varying dimple densities: non-textured (**a**), 15 dimples (**b**), 28 dimples (**c**), 45 dimples (**d**), 66 dimples (**e**), and 91 dimples (**f**).

Figure 8 presents the average pressure versus dimple number, and a linear increase in pressure is shown. The result indicates a high density of dimples causes higher pressure generation. Similar trends are also found by other researchers. Dobrica et al. [15] found the load carrying capacity of parallel sliders always increase with the dimple density using CFD. In addition, the trend of average pressure versus dimple density is in agreement with the review summarized in Gropper's review paper [2], where Gropper et al. summarized and pointed out that linear correlations between texture density and hydrodynamic lift generation are observed for both parallel and convergent contact. A possible explanation is that each dimple acts as a single wedge-like structure unit, which could generate additional hydrodynamic pressure independently. The higher pressure which results from increased dimple number could be considered as a collaborative effect of dimples. However, there should be a limitation for the dimple density considering texturing efficiency, wear, and other aspects. For example, dense dimples may result in stress concentration, weakening the surface, and thus lead to a high wear rate of the texture and component failure.

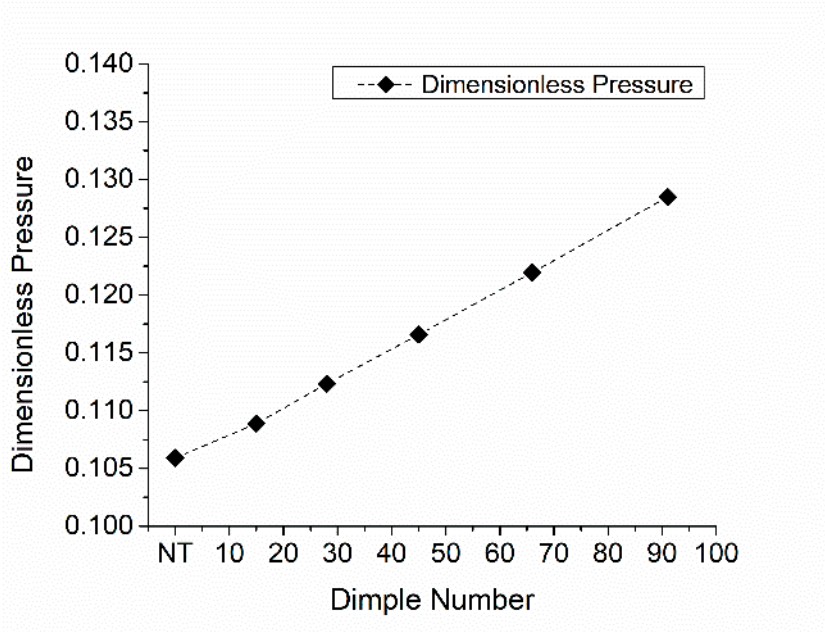

**Figure 8.** Average pressure on the top surface of the model for varying dimple densities (NT indicates the non-textured case).

### 4.3. Effect of Dimple Depth

Dimple depth is a vital factor for surface texture design because different dimple depths change the texturing processes, cost, and wear of the surface. Shallow dimples may wear out quickly, but deep dimples may negatively influence the strength of the part itself [16]. Choosing a proper dimple depth is important in applications of dimple texture. Simulation regarding the change dimple depth and film thickness are performed, the simulation condition is shown in Table 1 as Case 4.3.

Figure 9 shows the average pressure on the top surface of the model for each dimple depth and minimum film thickness $H_{min}$. When $H_{min}$ is 10 μm, the pressure increases with an increase of dimple depth until 10 μm; then, the pressure decreases as dimple depth becomes deeper. In addition, with a larger $H_{min}$, pressure increases due to the dimple texture being gradually minimized. When $H_{min}$ is 30 μm, the maximum pressure difference for the non-textured and textured cases is less than 3%. Papadopoulos et al. [9] analyzed the dimples effect on thrust bearing using CFD, the results indicates the optimized dimple depth close to the minimum film thickness. As shown in Figure 9, for all four lines, the highest pressure is shown when the dimple depth is 10 μm. Only for $H_{min}$ = 10 μm (black line)

is the optimized dimple depth close to the minimum film thickness. However, when $H_{min} = 20\ \mu m$ (blue line) or 15 μm (red line), a slight change of the pressure varying dimple depth from 5 to 20 μm is observed. In that case, the result of Figure 9 in agreement with Papadopoulos's conclusion when the $H_{min}$ is lower than 30 μm.

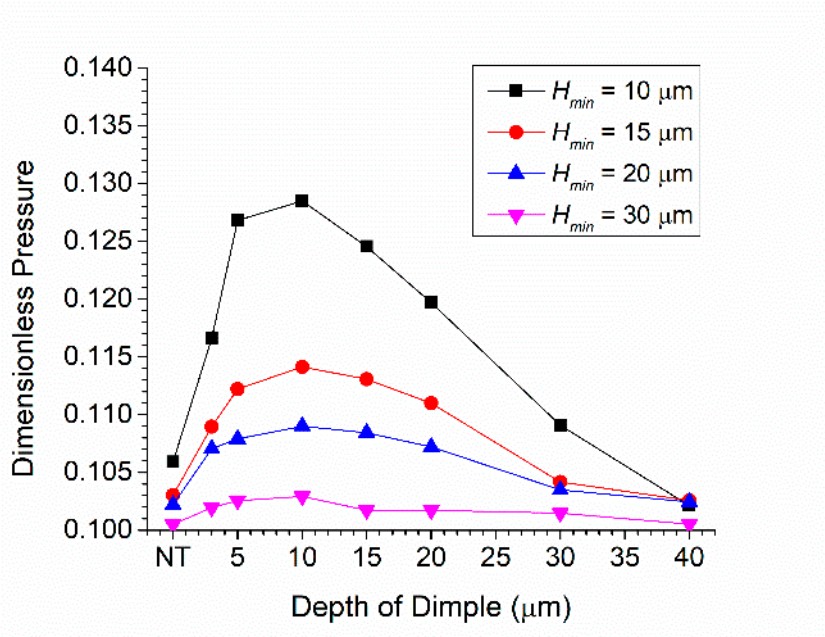

**Figure 9.** Comparison of average pressure on the top surface of the model for varying dimple depths (NT indicates the non-textured case).

A dimple depth of 10 μm causes increased pressure build-up and is deep enough to distinguish the texture from surface roughness. Further, the negative influence on strength is negligible for such a small depth compared to the entire bearing dimension.

In addition, it is difficult to maintain dimple depth variation to only a few microns by any texturing process currently. Further simulations are performed using the models and varying dimple depths in a single model. Models with random dimple depths in a single model are generated and distributed using MATLAB. Although the dimple depth for each dimple is modeled randomly, the depths are kept within a certain range and the average dimple depth for each single model is maintained at 10 μm exactly, as shown in Table 3. For additional pressure generation, the differences between the four cases is less than 1%, which is ignorable. The results suggest that, for dimple texture, an average depth may be used to evaluate texture quality.

**Table 3.** Comparison of different dimple depths.

| Case | Range of Depth | Average Depth | Dimensionless Pressure |
|------|----------------|---------------|------------------------|
| *a* | 10 μm | 10 μm | 0.128 |
| *b* | 9~11 μm | 10 μm | 0.128 |
| *c* | 7~13 μm | 10 μm | 0.128 |
| *d* | 5~15 μm | 10 μm | 0.127 |

### 4.4. Effect of Dimple Surface Angle

Dimple surface angle $\alpha$ is introduced to represent more accurately the geometric features of dimples using a simplified three-dimensional model in this study. The surface angle mimics a real dimple bottom surface profile with an acceptable simplification, which is easily utilized in numerical

studies to investigate the dimple surface texture effect. It is important to understand how this factor influences additional hydrodynamic pressure generation.

To investigate the influence of dimple surface angles, 9 sets of dimple surface angles varying from 90° to 7.5° are applied in the simulations, seeing Case 4.4 in Table 1. Figure 10 shows the pressure distributions on the bottom surface of the model for varying dimple surface angles. Pressure does not significantly change as the dimple surface angle decreases from 90° to 30°, at which point, the pressure drops rapidly as the dimple surface angle gets smaller.

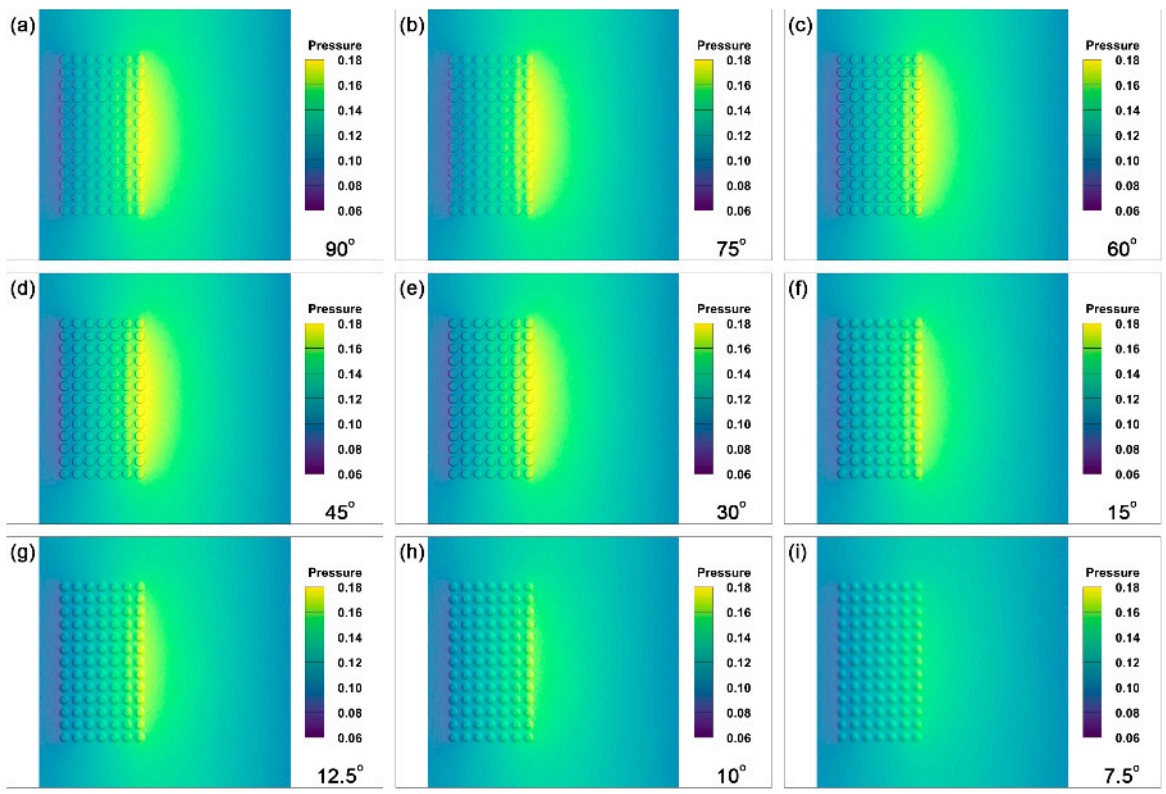

**Figure 10.** Pressure distributions on the bottom surface of the model for dimple surface angle 90° (a), 75° (b), 60° (c), 45° (d), 30° (e), 15° (f), 12.5° (g), 10° (h) and 7.5° (i).

The average pressure, dimple volume, and varying dimple surface angle are plotted in Figure 11. The changing trends of average pressure and dimple volume are similar: pressure decreases as dimple volume decreases. However, in the present study, change of depth could also lead to changes in dimple volume, but the trend of pressure change is opposite (Figure 7). In summary, when dimple size and dimple depth are fixed, reducing dimple volume by changing the dimple surface angle results in a decrease in hydrodynamic pressure generation.

However, it is noted that, the dimple surface angle can be changed from 90° to 30° without a significant reduction in hydrodynamic pressure generation. This result is important for texturing dimples using lasers and micromachining tools because there is no need to restrict the dimple bottom profile. In the laser texturing process, the dimple bottom profile may relate to the laser beam profile types, output power, workpiece material, etc.

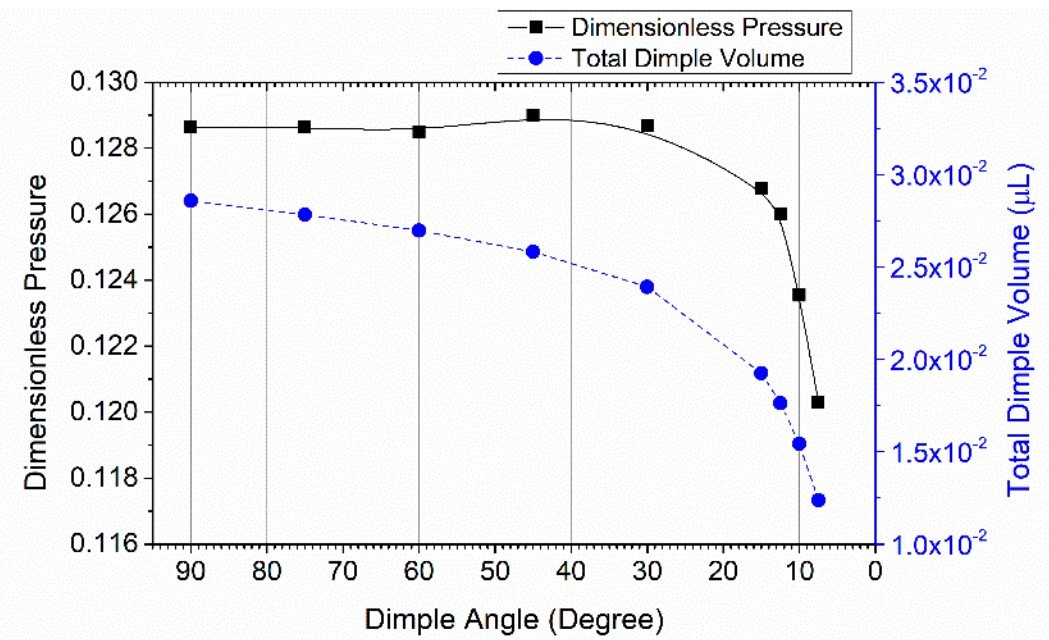

**Figure 11.** Average pressure on the top surface of the model and the total volume of dimples as the dimple surface angle changes.

There are several mechanisms that respond to the texture effect according to the literature, including the flow inertia effect [17], local cavitation [7] and collective dimple effect [18]. Tønder believes a step-like structure is formed by partial texture under fully lubricated contact conditions; the mechanism of texture effect is similar to the Rayleigh step bearing [19]. Each single dimple forms a mini wedge-like configuration that generates additional hydrodynamic pressure. By changing the geometric features of the dimples, including dimple shape and depth, the convergence ratio and effective wedge-like configuration area are changed as well, which results in changing the texture effect.

## 5. Conclusions

A numerical study on the effect of the geometric features of dimple textures on pressure build-up is performed in the present work. The contribution of the paper is the introduction of a new parameter called dimple surface angle, which is investigated to improve three-dimensional models that represent the real geometric features of dimples in a numerical study. The influences of the geometric features on texture effect, particularly the generation of additional hydrodynamic pressure, are investigated. The following conclusions are drawn from this study:

The dimple shape greatly influences the dimple texture effect. Squared dimples generate the highest additional pressure build-up. However, considering the feasibility of the texturing process, dimples with a circle shape have greater potential to be applied on a large scale.

- The dimple texture effect increases linearly as density increases.
- An optimal dimple depth is identified with a certain minimum film thickness. In this study, 10 μm is an optimal value when the minimum film thickness is 10, 15, 20, and 30 μm.
- As the minimum film thickness increases from 10 to 30 μm, the effect of dimple texture is minimized.
- The introduced dimple surface angle influences the dimple texture effect. The most significant changes occur when the dimple surface angle is smaller than $30^{\circ}$.

**Author Contributions:** Conceptualization, Y.W., R.T., and A.A.; methodology, Y.W. and R.T.; investigation, Y.W., R.T., and A.A.; original draft preparation, Y.W.; writing—review and editing, R.T. and A.A. All authors have read and agreed to the published version of the manuscript.

**Funding:** The publication was founded by VINNOVA Sweden's Innovations Agency under the scientific research project nr 2017-05540 and supported by centre for Design and Management of Manufacturing Systems DMMS at KTH Royal Institute of Technology.

**Conflicts of Interest:** The authors declare no conflict of interest.

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
