# Peer review of "Numerical Study of the Influence of Geometric Features of Dimple Texture on Hydrodynamic Pressure Generation"

_metals, doi:10.3390/met10030361_

Round 1

Reviewer 1 Report

Line 121 - The mass transfer mechanism between between ...

Line 128, 129 - formulas (4-5) and captions do not correlate (see attached file). The designations of the variables after the formulas should correspond to those used.

Line 138 - It is not clear from the text what pressure was considered: excessive or absolute?

Line 145 - The diameter of the dimple is not defined. Does it affect the hydrodynamic pressure in the layer?

Line 154 - The author states that dimples of any shape increase pressure. From a physical point of view, explanation is not given for this fact.

Author Response

We really appreciate the reviewer’s comment. valuable feedback. Please see the attachment.

Reviewer 2 Report

The authors report a study on the hydrodynamic pressure of different texturing geometries and dimples density.

This research paper is relevant and of interest to the audience of this journal. Nevertheless, there are some observations and mayor changes are recommended.

First of all the title do not provide an accurate idea of the work presented in the paper. This paper analyse the effect of the geometrical features on the hydrodynamic pressure, not on the friction neither on the wear behaviour. Both analysis were expected as “Surface Texturing Effect” For this reason, the authors should consider to change the title, being more specific about the developed study. Similarly, I strongly recommend to include more keywords.

1. Introduction:

Introduction could be improved and the bibliography used updated including the recent advanced of the last year.

2. Modelling dimples:

This section should be clearer. It should provide an overview of the performed analysis, specifying the changes on the geometrical features. Figure 4 should be included in this section, not in section 3. Figure 2 could be improved and the dimension annotation clarified, it is difficult to understand the system used, I recommend the authors to use a standard annotation method, or at least close to a standard one. It isn’t clear the distance between dimples used for the density effect study. Table 2 should be explained in section 2 not in section 4. Why

4. Results and discussion:

Data are not validated for any of the simulation so there is no reference for simulation error. There is little result discussion or comparison with other authors results, when it is, the other authors are not referred. Why is dimensional pressure linear to dimple number? Is there an explanation?  Additionally some aesthetic improvements should be done. Figure 5 and 10 provide a double identification of the subfigures but they are not referred in the caption. Scale of figure 7 is too small and subfigures are not identified.

Author Response

(The authors gave the same response as above.)

Round 2

Reviewer 2 Report

Figure 2 was not modified, it is still dificult to understand, labels for figure 5 are still double and codification style (Td, Td-R) is not defined until figure 6, which is confusing.

Refence is not included in Line 258 after "Dobrica et al." and in line 287 after "Papadoulos et al." please check there are no more cases in the hole text.

Author Response

We thank the reviewer for valuable feedback. Please see the attachment.
